# In Vivo Biodistribution, Clearance, and Biocompatibility of Multiple Carbon Dots Containing Nanoparticles for Biomedical Application

**DOI:** 10.3390/pharmaceutics13111872

**Published:** 2021-11-05

**Authors:** Jinfeng Liao, Yuan Yao, Cheng-Hao Lee, Yongzhi Wu, Pei Li

**Affiliations:** 1State Key Laboratory of Oral Diseases, National Clinical Research Center for Oral Diseases, West China Hospital of Stomatology, Sichuan University, No. 14, Section 3, Southern Renmin Road, Chengdu 610041, China; liaojinfeng.762@163.com (J.L.); wuyongzhiscu@163.com (Y.W.); 2Department of Applied Biology and Chemical Technology, The Hong Kong Polytechnic University, Hung Hom, Kowloon, Hong Kong, China; yychips@gmail.com (Y.Y.); chenghao.lee@polyu.edu.hk (C.-H.L.)

**Keywords:** multiple carbon-dot nanoparticle, single carbon dot, biodistribution, clearance, biocompatibility

## Abstract

Current research on the use of carbon dots for various biological systems mainly focuses on the single carbon dots, while particles that contain multiple carbon dots have scarcely been investigated. Here, we assessed multiple carbon dots-crosslinked polyethyleneimine nanoparticles (CDs@PEI) for their in vivo biodistribution, clearance, biocompatibility, and cellular uptake. The in vivo studies demonstrate three unique features of the CDs@PEI nanoparticles: (1) the nanoparticles possess tumor-targeting ability with steady and prolonged retention time in the tumor region. (2) The nanoparticles show hepatobiliary excretion and are clear from the intestine in feces. (3) The nanoparticles have much better biocompatibility than the polyethyleneimine passivated single carbon dots (PEI-CD). We also found that pegylated CDs@PEI nanoparticles can be effectively taken up by the cells, which the confocal laser scanning microscope can image under different excitation wavelengths (at 405, 488, and 800 nm). These prior studies provide invaluable information and new opportunities for this new type of intrinsic photoluminescence nanoparticles in carbon dot-based biomedical applications.

## 1. Introduction

Carbon dots (CDs) are emerging nanomaterials for nanomedicine due to their photoluminescence property with tunable emission possibility, nanometer size (a few nanometers), surface functionality, excellent physicochemical and photochemical stabilities, good biocompatibility, and low cytotoxicity [1,2,3,4,5,6]. These unique features have led to much research development in bioimaging, drug delivery, cancer theranostics, photodynamic therapy, gene therapy, antimicrobial agents, and biosensing [7,8,9,10,11,12,13]. Various synthetic strategies have been developed for the preparation of passivated single CD nanoparticles of a few nanometers [14,15,16,17,18,19,20,21,22,23], and in vivo fates of the CD nanoparticles have also been investigated for clinical translation. Chen et al. prepared near-infrared fluorescent CD nanoparticles to track their in vivo behaviors. The animal experiment showed that CD nanoparticles (average diameter = 4.1 nm) were concentrated in the kidney at one-hour post-injection. There were no significant signals in any organs after 24 h. These results suggested that the CD nanoparticles were cleared mainly through the kidney via urinary excretion, not by the liver or spleen. Tumor-to-background contrast was also found to have enhanced, suggesting the passive tumor targeting of the CD nanoparticles [24]. Xie et al. investigated the clearance of injected Gd-encapsulated carbonaceous dots (Gd@CDs) with in vivo MRI and found that signal decay started at 1 h after injection. At the time point of 4 h post-injection, signals in major organs had almost decreased to the original level. Their results also confirmed efficient renal clearance of CD nanoparticles evidenced by strong signal enhancement in the bladder [25]. In contrast to the extensive studies of single CD systems, there are only a few reports on the fabrication of particles containing multiple CDs. Liu et al. reported the preparation of carbon dots encapsulated in mannosylated liposomes (CDs@liposomes) for targeting recognition and tracking of interaction between carbohydrates and glycoprotein [26]. They found that the CDs in liposomes showed photoluminescence intensity five times higher than that of unencapsulated CDs, which may attribute to the organic groups of liposomes that passivated the surface of loaded CDs. The enhanced photoluminescence emission of the CDs@liposomes improved tracking of the anticancer drug in tumor therapy. In another study, nanoaggregates of carbon dots were fabricated through electrostatic interaction and self-assembled carbon dots loaded with anticancer drugs for tumor cell targeting image and drug delivery [27]. To the best of our knowledge, there have not yet been any studies on in vivo biodistribution, clearance, and cytotoxicity of particles containing multiple carbon dots.

We have recently developed a synthetic strategy to prepare a new type of multiple carbon dot-crosslinked polyethyleneimine particles (CDs@PEI) [28]. The particles are prepared by a two-stage reaction: the methyl methacrylate (MMA) swollen PEI-g-PMMA micellar nanoparticles are first synthesized, followed by hydrothermal treatment of the nanoparticles to generate CDs@PEI particles. The obtained particles contain numerous CD nanoparticles, which are homogeneously embedded within the PEI network. The CDs@PEI nanoparticles possess a larger diameter (about 100 nm) than that of individual CD nanoparticles (<10 nm in diameter). They also possess a high quantum yield of up to 66%, excitation-dependent emission, as well as pH- and photo-stable photoluminescence. In this work, we studied the biodistribution, clearance, and systemic biocompatibility of the CDs@PEI nanoparticles for their potential biomedical application (Figure 1). The biodistribution and clearance of the nanoparticles were investigated through qualitative and semi-quantitative measurements of photoluminescence emission in 4T1 tumor-bearing BALB/c mice. The fluorescent images of ex vivo organs at different treatment times indicated that intravenously injected CDs@PEI nanoparticles were much more effective in accumulating in tumor sites and had prolonged and steady tumor retention properties when compared with single PEI-CD nanoparticles. Such prolonged tumor uptake is highly desirable in cancer nanomedicine [29,30,31,32]. The CDs@PEI nanoparticles were also found to be mainly excreted from the hepatic route, which was different from the single PEI-CD nanoparticles that were excreted from the renal system [24]. Furthermore, the CDs@PEI particles had much lower systemic toxicity than the single PEI-CD nanoparticles. They showed no detectable pathological damages such as cell necrosis, cell apoptosis, and inflammation in the major organs. We have also exploited potential applications of CDs@PEI nanoparticles for multi-color fluorescence cell imaging. Our studies have demonstrated that the multiple carbon dots-crosslinked PEI nanoparticles represent a new design of carbon dot-based particles as an intrinsic photoluminescent nanocarrier for tumor targeting and safe tumor theranostics.

## 2. Materials and Methods

### 2.1. Reagents, Cell Lines, and Animals

Methyl methacrylate (MMA), branched polyethyleneimine (PEI, Mw = 25 kDa), and *tert*-butyl hydroperoxide (TBHP, 70%) were purchased from Sigma-Aldrich (St. Louis, MO, USA). Methoxy poly(ethylene glycol) succinimidyl propionate (mPEG-SPA, Mw 5 kDa, from Bio-Carrier International Ltd., Hong Kong, China) was used as received. Dulbecco’s modified Eagle’s medium (DMEM, high glucose), fetal bovine serum (FBS, E.U. approved, South America origin), and penicillin/streptomycin (P/S, 10,000 U/mL) were all purchased from ThermoFisher Scientific (Waltham, MA, USA). Human cervix adenocarcinoma (HeLa) cells were obtained from the American Type Culture Collection. BALB/c mice (female, 6–8 weeks old) were obtained from Beijing HFK Bioscience Co. Ltd., Beijing, China. The animal procedures were performed in compliance with the Institutional Animal Care and Ethics Committee of Sichuan University (Chengdu, China) and carried out according to the Guidelines on Animal Care of Sichuan University (The approval code is WCHSIRB-D-2019-074, and the date is 28 February 2019).

### 2.2. Synthesis and Characterization of CDs@PEI and PEI-CD Nanoparticles

The CDs@PEI and PEI-CD nanoparticles were synthesized by hydrothermal treatment at 150 °C for 48 h according to a previously established method of our group [28]. The final product was purified via dialysis in deionized water for 3 days until the conductivity of the dialysate was below 10 μS/cm. The particle sizes and *ζ*-potential values of the CDs@PEI nanoparticles were determined by a Delsa^TM^Nano Particle Analyzer (Beckman Coulter, Brea, CA, USA). These values were obtained according to the average value of three repeated measurements with a standard deviation. The morphology of the nanoparticles was observed by a transmission electron microscope (JEOL, Tokyo, Japan, 100 CXII TEM) at an accelerating voltage of 100 kV. The sample was prepared by wetting a carbon-coated grid with a small drop of dilute particle dispersion (10 μL, 50 mg/mL). Upon drying, the sample was stained with 0.5 *w*/*w*% phosphotungstic acid (PTA, WO_2_·H_3_PO_4_·xH_2_O) solution for 1 min, then dried at room temperature before analysis. All TEM images were taken in bright field mode. The photoluminescent (PL) spectra of the samples were recorded with a HORIBA Scientific FluoroMax-4 spectrofluorometer (Horiba Jobin Yvon, Edison, NJ, USA) from 300 to 480 nm.

### 2.3. PEGylation of CDs@PEI Nanoparticles

The mPEG-SPA stock solution (1 mg/mL) was freshly prepared by adding 10 mg of mPEG-SPA into 10 mL of DI water. The CDs@PEI nanoparticle dispersion (5 mL, 1 mg/mL) was added to 5 mL of PBS solution, then mixed with 2.5 mL mPEG-SPA stock solution. The mixture was stirred at room temperature for 12 h. The weight ratio of particle to PEG was at 2 to 1. The sample was further purified by dialysis in DI water for two days.

### 2.4. Cell Imaging Using the Confocal Laser Scanning Microscopy

The Leica-SP8-MP confocal laser scanning microscopy (CLSM, Leica Microsystems, Wetzlar, Germany) was used for the cell imaging experiment. Hela cells were seeded in a 35 mm petri dish containing 2 × 10^5^ cells per dish. The sample (20 μg/mL) was mixed in a culture medium and co-incubated with Hela cells for 24 h at 37 °C in a humidified atmosphere containing 5% CO_2_. The culture medium was then removed, followed by rinsing the cells twice with phosphate-buffered saline (PBS) solution. Finally, an additional 1 mL of the PBS solution was added for scanning. For downconversion cell imaging, both lasers of 405 and 488 nm wavelengths were chosen, while for upconversion cell imaging, an 800 nm pulse laser was used.

### 2.5. In Vivo Biodistribution and Clearance of CDs@PEI Nanoparticles

One Million 4T1 tumor cells, dispersing in a cell culture medium without serum (100 μL), were injected into the subcutaneous region in each mouse to induce solid tumors. After one week, the tumor diameter in the right rear flank of the mice was about 5 mm. Subsequently, the tumor-bearing mice were administered with CDs@PEI (5 mg/kg) nanoparticle dispersion through tail vein injection. At 0.5, 1, 4, 24, and 30 h after injection, the ex vivo major organs and tumor tissue were removed and fluorescently imaged (Ex = 500 nm, Em = 620 nm) using an IVIS Lumina (Perkin-Elmer) at various time intervals. The organ regions were selected as the region of interest (ROI) for quantitative analysis of fluorescence intensity. We also removed the stomach and intestine to study the clearance behaviors of the nanoparticles.

### 2.6. Systemic Toxicity of the CDs@PEI Nanoparticles

Female BALB/c mice were allowed to adapt to the environment for 1–2 weeks. When the mice grew to a weight of around 20 g, they were administered with CDs@PEI (5 mg/kg, 100 µg nanoparticles injected to mice) through tail vein injection. The control group was given normal saline. Fourteen days after the injection, the mice were sacrificed, and their major organs were harvested for routine hematoxylin and eosin (H&E) staining.

## 3. Results and Discussion

### 3.1. Preparation and Characterization of CDs@PEI Nanoparticles

The CDs@PEI nanoparticles used in this study were prepared according to our previously established method which involved the synthesis of MMA swollen PEI-g-PMMA micellar nanoparticles, followed by hydrothermal treatment of the nanoparticles for 48 h to generate in situ carbon dots. The resulting CDs@PEI nanoparticles had an average particle size of around 100 nm (Figure 2A) and the *zeta*-potential value was around +35 mV at pH 7. The particle sizes of CDs@PEI nanoparticles were at least 10 times larger than that of single carbon dots ranging from 4 to 10 nm. Figure 2B illustrates the particle morphology of the CDs@PEI nanoparticles, which contain numerous carbon dots with irregular shapes. Figure 2C displays the appearance of CDs@PEI solution under visible light and irradiation with 365 nm UV light. Figure 2D shows the photoluminescence spectra of CDs@PEI nanoparticles under excitation wavelengths from 300 to 480 nm. The strongest emission peak centered at 470 nm upon the excitation at 340 nm. The emission wavelength was not sensitive to the excitation wavelength in the excitation range of 300 to 360 nm. Further increasing the excitation wavelengths from 380 to 480 nm shifted the emission peaks to longer wavelengths to 520 nm with reduced intensities. The CDs@PEI nanoparticles were also found to have excellent physical and chemical stabilities. Figure 3 shows that strong fluorescence intensities remained in a wide pH range (pH 1.2, 2.8, 4.7, 6.6, 8.4, and 10.3) under a 365 nm UV excitation. Moreover, the particle sizes had almost no changes between pH 1 and 8 and are slightly reduced with pH higher than 8.

### 3.2. Cell Imaging of CDs@PEI Nanoparticles

Our previous study demonstrates that the CDs@PEI nanoparticles offer several advantages such as high fluorescence quantum yield in an aqueous system, excitation-dependent emission, and excellent pH- and photo-stability. Thus, we evaluated the potential application of the nanoparticles as fluorescence imaging probes for cell imaging. The CDs@PEI nanoparticles were first modified with mPEG-SPA in a weight ratio of 2:1 to enhance their stability in the culture medium. After purification of the modified nanoparticles via dialysis in DI water, the average hydrodynamic diameter of the PEGylated CDs@PEI nanoparticles was 109 nm, which is slightly larger than the CDs@PEI nanoparticles (99.5 nm in diameter). Furthermore, the surface charge of the PEGylated CDs@PEI nanoparticles was reduced from +34 mV to +24 mV. The increase of particle size and decrease of the *zeta*-potential value of the PEG-CDs@PEI nanoparticles indicated the successful modification of the CDs@PEI nanoparticles to contain nonionic methoxy poly(ethylene glycol) groups on the particle surface. The fluorescence imaging performance was evaluated by a CLSM system. HeLa cells were incubated with PEG-CDs@PEI nanoparticles (20 µg/mL) for 24 h, followed by rinsing the cells three times with PBS solution. Figure 4 exhibits images of HeLa cells observed under a bright field and two different exciting wavelengths (405 and 488 nm). Both the cell membrane and the cytoplasm are brightly illuminated with blue and yellow under the microscope. In addition, the nanoparticles are mainly distributed in the cytoplasm, and none of them were observed inside the nucleus.

The upconversion cell image was also observed by the confocal fluorescence microscope under the excitation of an 800 nm pulse laser (Figure 5). A bright upconversion fluorescence from the sample was observed (to distinguish the upconversion image from the downconversion images, green was selected to represent the color of fluorescence). A brightly green fluorescence image of the HeLa cells with the cell membrane and the cytoplasm was clearly labeled. This feature of near-infrared excitation wavelength of the CDs@PEI nanoparticles offers the additional advantage of low photodamage to biological samples. These results demonstrate that the PEG-CDs@PEI nanoparticles can be effectively taken up by the cells and they possess multicolor fluorescence properties for in-vitro cell imaging.

### 3.3. In Vivo Biodistribution of CDs@PEI Nanoparticles

The in vivo biodistribution and systemic toxicity of the CDs@PEI nanoparticles were evaluated for biomedical application as fluorescence imaging nanocarrier candidates for tumor theranostics. The CDs@PEI nanoparticles had an average particle size of approximately 100 nm, which was different from that of the single carbon dot of PEI-CD (<10 nm). BALB/c mice with 4T1 tumors were used as an animal model. In the biodistribution study, CDs@PEI nanoparticles at a dosage of 5 mg/kg were administered to the model mice through tail vein injection once their tumors had grown to the size of approximately 5 mm. The tumor-bearing mice were sacrificed after post-injection of samples at each time interval, and their major organs and tumors were then isolated. Figure 6A shows that the CDs@PEI nanoparticles mainly accumulated in the tissues of the liver, kidneys, and tumor. ROI analysis results shown in Figure 6B indicate that the fluorescence intensity in the major organs and tumor was in the sequence of liver > kidney > tumor > lung > spleen > heart at the early stages (30 min, 1 h, and 4 h). However, the order changed to tumor > liver > kidney > lung > spleen > heart at 30 h. These results indicate that the CDs@PEI nanoparticles accumulated in the tumor site and underwent hepatic clearance from the body [33,34,35]. Moreover, the average fluorescence intensity of the CDs@PEI nanoparticles at the tumor site rapidly increased within 0.5 h post-injection and remained steady at 4–30 h after injection (Figure 6C). These in vivo biodistribution results were quite different from those of single carbon dots. For example, Wang’s group reported that single carbon dots took 5 h to achieve maximum accumulation at the tumor site, and the fluorescence intensity of carbon dots steadily decreased over time [36]. Lee et al. also reported that fluorescence signals of carbon dots could not be detected in the tumor at 24 h after injection [37]. In our case, the CDs@PEI nanoparticles had a fast accumulation (within 30 min) as well as a long and steady retention time at the tumor site. There were two plausible reasons for the enhanced retention: (1) the enhanced permeability and retention (EPR) effect in nanoparticles with larger particle size (CDs@PEI) might be the main cause [38]. (2) The acidic extracellular pH of tumor tissues might trigger the proton-sponge effect of the PEI, thus enhancing the cellular uptake of nanoparticles into the tumor cells via endocytosis [39]. This effect is evident by the above-mentioned *in vitro* cellular uptake results.

### 3.4. In Vivo Clearance of CDs@PEI Nanoparticles

To further study the clearance route of the CDs@PEI nanoparticles, the stomach and intestine of the treated mice were isolated and examined with the ex vivo fluorescent images (Figure 7). Strong fluorescence signals were observed from the intestine at 24 h after injection of the CDs@PEI nanoparticles. At 30 h, the signal intensity significantly reduced, which may attribute to the clearance of the CDs@PEI nanoparticles via feces. Therefore, the results of the clearance route suggest that the CDs@PEI nanoparticles were mainly excreted from the intestine in the feces. In addition, the images of the liver shown in Figure 6A also illustrate strong fluorescence signals, indicating the hepatobiliary excretion of the CDs@PEI nanomaterials. It has been reported that glomerular filtration by the kidneys is highly size-selective [40,41]. Previous in vivo biodistribution studies of single carbon dots with a diameter less than 10 nm proved that the kidneys were the main clearance organ [42,43,44]. Our results indicate that the CDs@PEI nanoparticles with approximately 100 nm in diameter could be excreted from the intestine in the feces.

### 3.5. In Vivo Biocompatibility of the CDs@PEI Nanoparticles

The in vivo biocompatibilities of the PEI-CD dots and CDs@PEI nanoparticles were evaluated using the BALB/c mice with normal saline-injected mice as the control group. The cytotoxicity of the PEI-CD dots was found to be much higher than that of the CDs@PEI nanoparticles. In our experiment, female BALB/c mice (n = 5) were administered with PEI-CD dots (5 mg/kg, i.e., 100 μg single carbon dots) through tail vein injection. All the mice died within 15 min, which might be due to the high toxicity of free PEI. In comparison, the survival rate of the mice injected with 5 mg/kg CDs@PEI nanoparticles was 100%. The mice appeared normal like the control group, throughout the whole study. Two weeks after injecting the CDs@PEI nanoparticles, the mice were sacrificed and H&E staining was carried out in ex vivo major organs. Figure 8 shows that there were no detectable pathological damages such as cell necrosis, cell apoptosis, and inflammation in the major organs for the CDs@PEI group when compared with the control group. The results suggest that the CDs@PEI nanoparticles had much lower systemic toxicity than the PEI-CD dots. Therefore, the CDs@PEI nanoparticles are promising photoluminescent nanocarriers for concurrent diagnosis and treatment of cancer with a trackable delivery process and biodistribution.

## 4. Conclusions

In this study, fluorescent nanoparticles containing multiple carbon dots (CDs@PEI nanoparticles) were prepared and evaluated for their potential biomedical application. The pegylated CDs@PEI nanoparticles can be effectively taken up by the cells, which the confocal laser scanning microscope can image under different excitation wavelengths (at 405, 488, and 800 nm). In vivo biodistribution, clearance, and biocompatibility were evaluated systematically. Three unique features of the CDs@PEI nanoparticles were found through the in vivo studies. The nanoparticles possessed a steady and prolonged retention time at the tumor region. They could be excreted from the hepatobiliary system and cleared from the intestine in feces. Furthermore, the CDs@PEI nanoparticles possessed good biocompatibility and were much less toxic than the single PEI-CD dots. Our results show that the CDs@PEI nanoparticles, as a new kind of multiple carbon dots containing nanoparticles, are promising photoluminescent nanomaterials for tumor theranostics.

## Figures and Tables

**Figure 1 pharmaceutics-13-01872-f001:**
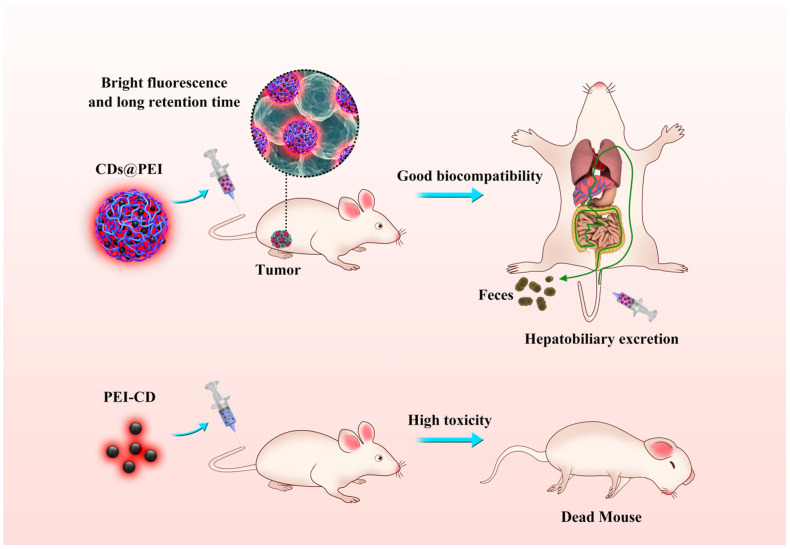
Top: The biodistribution, clearance, and systemic biocompatibility of CDs@PEI nanoparticles in mice; bottom: PEI-CD single carbon dots showed high toxicity in the animal.

**Figure 2 pharmaceutics-13-01872-f002:**
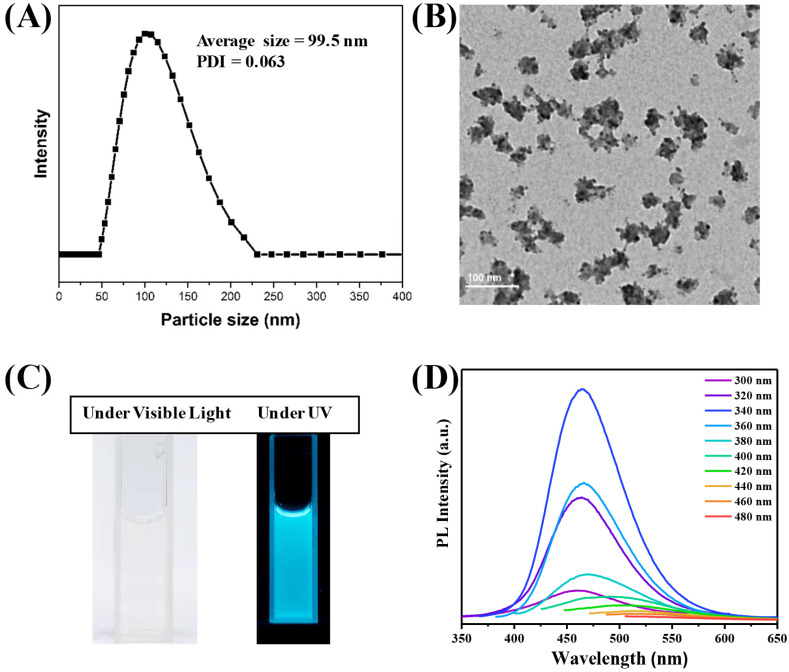
(**A**) Hydrodynamic size and distribution of the nanoparticles measured by the dynamic light scattering in water. (**B**) TEM image of morphology of CDs@PEI nanoparticles. (**C**) Photos of CDs@PEI solutions under visible and UV lights (irradiation with 365 nm UV light). (**D**) Photoluminescence spectra of CDs@PEI nanoparticles under excitation wavelengths from 300 nm to 480 nm.

**Figure 3 pharmaceutics-13-01872-f003:**
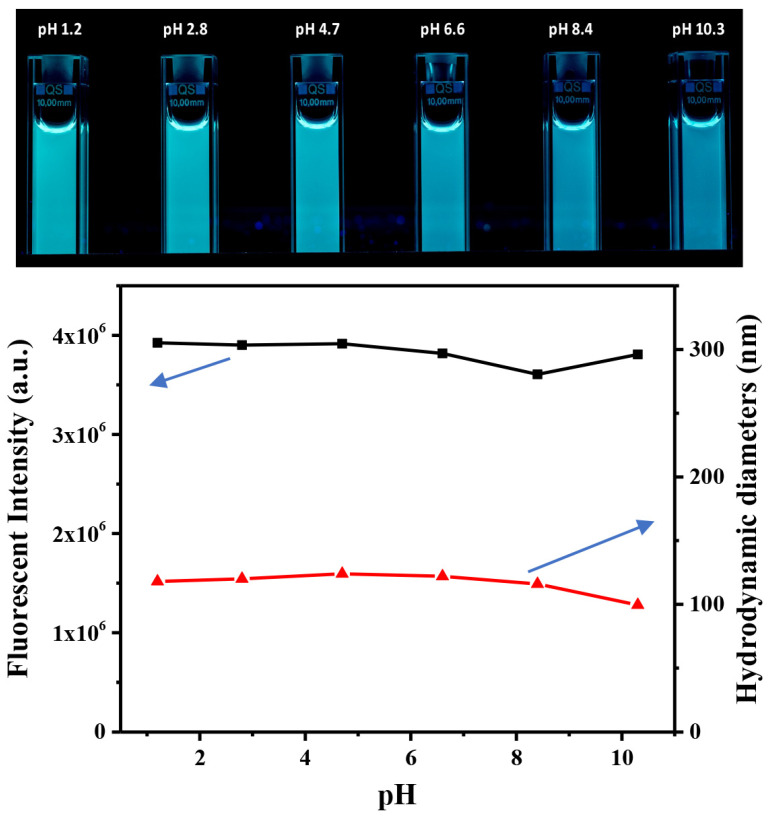
Top: Appearance of CDs@PEI solutions at pHs between 1.2 and 10.3 under UV light (irradiation with 365 nm UV light); bottom: the fluorescent intensities (blank line) and particle sizes (red line) of CDs@PEI nanoparticles at different solution pHs.

**Figure 4 pharmaceutics-13-01872-f004:**
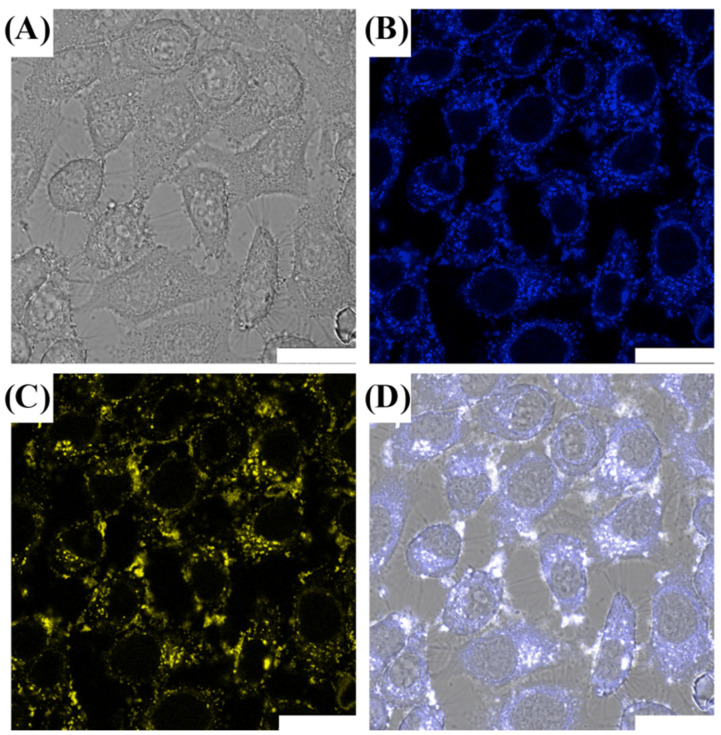
Laser scanning confocal microscopy images of PEG-CDs@PEI incubated with HeLa cells for 24 h. The cells were observed (**A**) under bright field; (**B**) excited at 405 nm; (**C**) excited at 488 nm; (**D**) overlapped images of B and C (scale bar = 25 μm).

**Figure 5 pharmaceutics-13-01872-f005:**
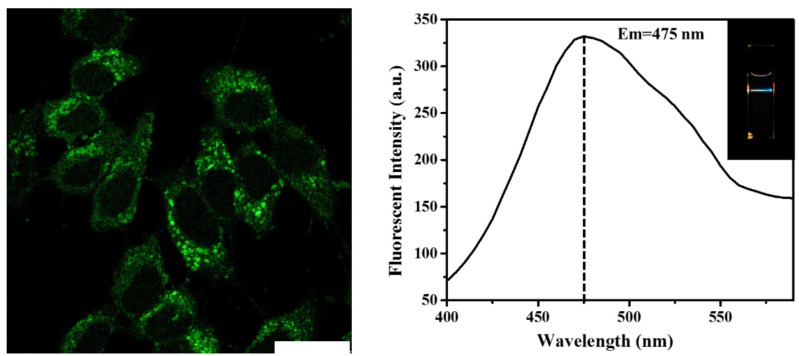
Laser scanning confocal microscopy image of PEG-CDs@PEI nanoparticles incubated with Hela cells for 24 h (Scale bar = 25 μm). The sample was observed under the excitation of an 800 nm pulse laser.

**Figure 6 pharmaceutics-13-01872-f006:**
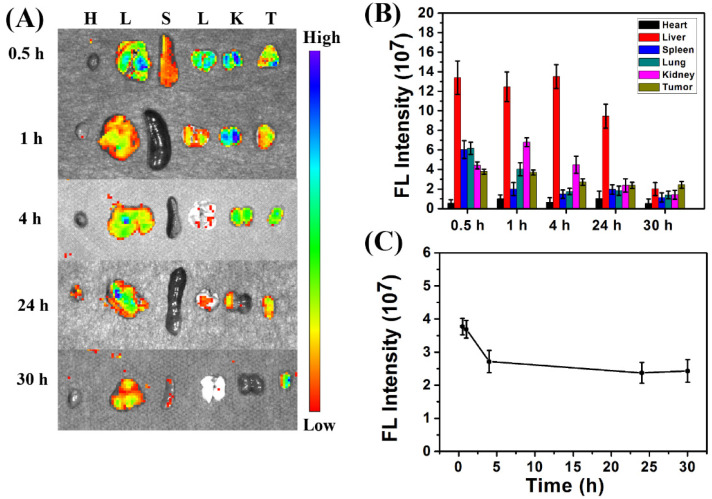
(**A**) Fluorescence image of ex vivo tissues (H, heart; L, liver; S, spleen; L, lung; K, kidney; T, tumor) isolated from mice after injection of CDs@PEI nanoparticles at 0.5, 1, 4, 24, and 30 h. (**B**) Average fluorescence (FL) intensities of major tissues and tumor at the designated time interval. (**C**) FL intensity curve of the tumor with specific time point.

**Figure 7 pharmaceutics-13-01872-f007:**
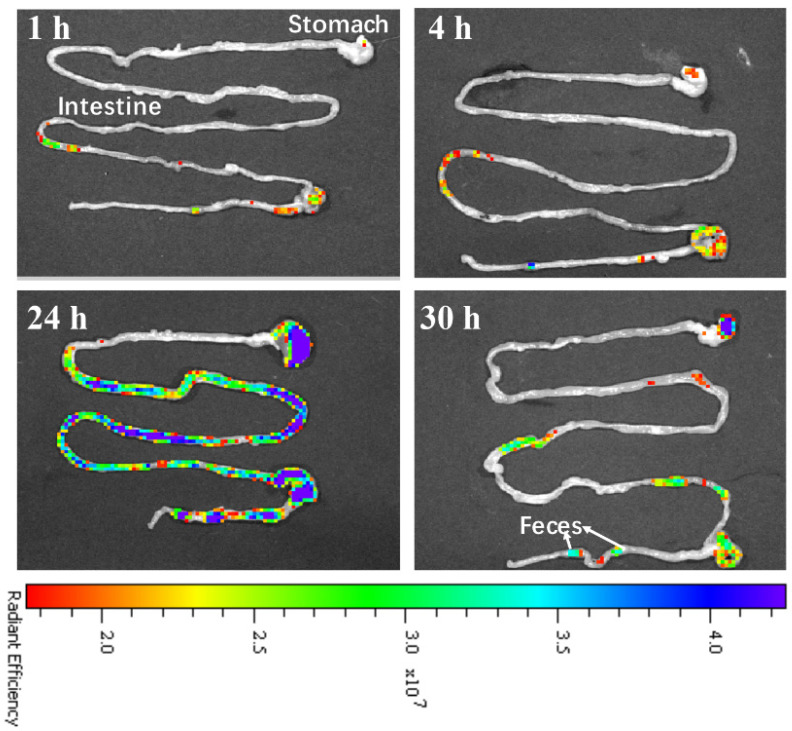
Fluorescent images of the ex vivo stomach and intestine from mice injected with CDs@PEI nanoparticles at 1, 4, 24, and 30 h.

**Figure 8 pharmaceutics-13-01872-f008:**
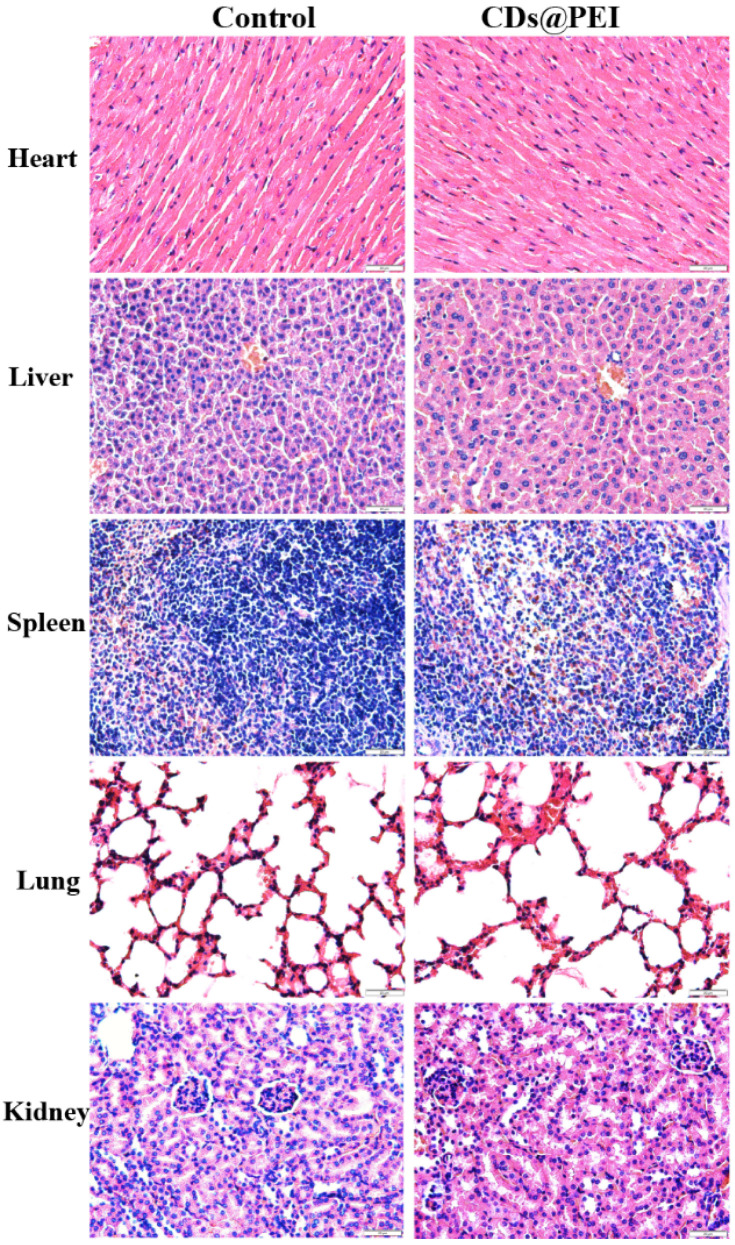
H&E staining from the major tissues of BALB/c mice at two weeks after injection with saline and CDs@PEI nanoparticles (scale bar = 20 μm).

## Data Availability

Not applicable.

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
