# Peer review of "In Vivo Biodistribution, Clearance, and Biocompatibility of Multiple Carbon Dots Containing Nanoparticles for Biomedical Application"

_pharmaceutics, 2021, doi:10.3390/pharmaceutics13111872_

Round 1

Reviewer 1 Report

The concept of carbon dots (CDs) development for biological and biomedical applications mostly expired. Several limitations make it unsuitable for biological applications. One of the main reasons is its photoresponsive is belong to ultraviolet (UV) light (200-400 nm) which is harmful to cells. Several attempts are in progress to make it visible and NIR responsive including, up-conversion nanoparticles. Additionally, the biodistribution study for a new class nanomaterial is always essential; however, the distribution study of CDs (or CD@PEI) at this current time had already been rectified. The authors could focus their work differently where others did not notice up to now. It would increase the impact of this manuscript. The selective comments are the following:

  1. In the abstract, the reviewer does not agree to this claim of “while particles that contain multiple carbon dots have scarcely been investigated.” The authors can find huge published research: ACS nano. 2013 Jul 23;7(7):5684-93.; Small. 2012 Jan 23;8(2):281-90.; Int J Nanomedicine. 2020;15:6519-6529. The authors have reorganized this line.
  2. In the Figure 1 schematic diagram, the meaning of the lower part is not clear. The author can add a description in the Figure caption.
  3. Most of the nanoparticles, including, CDs have different size distributions. Therefore the size distributions measured by DLS have to add.
  4. The physical and chemical stability of the CDs@PEI nanoparticles needs to explore.
  5. In Figure 2b, the authors have to add the photo of the CDs@PEI nanoparticle under visible light.
  6. In Figure 3, the authors should add a merged image of 405 nm, 488 nm, and 800 nm excitation wavelength. The scale bar contrast needs improvement.

Reviewer 2 Report

In this work, large NPs of multiple C Dots incorporated into a matrix of PEI-g-PMMA crosslinked micellar nanoparticles, were tested for their in vivo biodistributiom, biocompatibility, excretion. Remarcably, these NPs showed different biodistribution properties as compared to single C Dots, in particular by steadily accumulating in tumor for at least 30h.

However, albeit of some interest, the results described are not sufficient to merit full approval of the manuscript, being at best considered as a preliminary work. Further experiments on possible therapeutic applications are indeed necessary.

In their conclusions, the authors anticipate potential applications as nanocarrier for tumor targeting and safe tumor theranostic. 

In particular, cationic polymer polyethylenimine could be exploited as a gene delivery system, e.g. siRNAs.
The efficacy could be compared to that of previously reported rPEI-CQDs (Liu et al., 2012; Wu et al., 2016).

Other opportunities could be the functionalization of amino groups for linking anti-cancer drugs by for instance an acid-labile Schiff base linkage (Jia et al., 2016).

Minor concerns

The size of the NPs should be analyzed by dynamic light scattering. The results should be correlated to the images obtained by TEM, to confirm homogeneous entrapment of the C Dots into the PEI- PMMA matrix.

Figure 2A, particle morphology, specify the experiment. 

Page 6, for clarity, rewrite the sentences: At 30 h, the signal intensity…….via feces.

Reviewer 3 Report

The authors report a study on the biodistribution, clearance, and systemic biocompatibility of previously synthesized multiple carbon dots cross-linked with polyethyleneimine and compared these properties with those of PEI passivated single carbon dots. The reported results have relevance in Nanomedicine field. I believe that this manuscript can be accepted for publication in this Journal after some revisions.

In particular, no characterization of the newly synthesized CDs@PEI nanoparticles modified with mPEG-SPA is reported; I would suggest the authors to confirm the effectiveness of pegylation reaction by FTIR spectroscopy and also to evaluate the particle size of the pegylated nanomaterials by TEM or DLS measurements.

Round 2

Reviewer 1 Report

The editor could accept the manuscript.

Reviewer 3 Report

I believe that this revised version of the manuscript can be accepted for publication in this Journal.